# Augmented Rehabilitation Program for Patients 60 Years and Younger Following Total Hip Arthroplasty—Feasibility Study

**DOI:** 10.3390/healthcare10071274

**Published:** 2022-07-09

**Authors:** Ahmed M. Negm, Milad Yavarai, Gian S. Jhangri, Robert Haennel, C. Allyson Jones

**Affiliations:** 1Department of Physical Therapy, University of Alberta, Edmonton, AB T6G 2G4, Canada; anegm@ualberta.ca (A.M.N.); myavari@ualberta.ca (M.Y.); bob.haennel@ualberta.ca (R.H.); 2School of Public Health, University of Alberta, Edmonton, AB T6G 1C9, Canada; gian.jhangri@ualberta.ca

**Keywords:** hip arthroplasty, rehabilitation, exercise, complex intervention

## Abstract

The purpose of this study is to examine the feasibility, safety and outcomes of a study comparing a 6-week post-operative rehabilitation program to usual care in patients ≤60 years undergoing elective unilateral total hip arthroplasty (THA). Methods: A cohort of 24 THA patients were recruited during their 6-week postoperative visit to their surgeons. The community-based rehabilitation program, which was designed to improve function and increase activity, consisted of 12 structured exercise classes on land and water over 6 weeks. Physical activity was assessed using a Sense Wear Pro Armband (SWA). Participants completed the Hip Osteoarthritis Outcome Score (HOOS) and THA satisfaction questionnaire before and immediately after the intervention. Results: 14 participants received the augmented rehabilitation, and 10 participants were in the control group. All except one in the intervention group completed at least 80% of the sessions. The intervention group took significantly more steps/day (mean difference = 2440 steps/day, 95% CI = 1678, 4712) (*p* < 0.05), at the follow-up compared to baseline. The intervention group had a higher mean change of number of weekly PA bouts than the control group. Within the intervention group, all HOOS subscales were significantly higher at the follow-up compared to baseline. Conclusion: Findings provided pragmatic insight regarding the intervention and assessments of implementing an augmented rehabilitation program for elective THA.

## 1. Introduction

In 2017–2018, approximately 53,217 elective total hip arthroplasty (THA) procedures were performed in Canada, representing a 17.4% increase over the past 5 years [1]. Of the hip arthroplasty hospitalizations, 18,920 (32.3%) were under the age of 65 and an increased number of younger patients are now undergoing hip arthroplasty. By 2030, 52% of primary THAs are projected to be implanted in patients younger than 65 years, with the greatest increase in patients aged 45–55 years [2,3]. This increase of THA in younger patients may play a role in the future rehabilitation demands for primary and revision THA surgery [4].

Osteoarthritis (OA) is the most common reason for patients older than 60 years of age to undergo THA [5]. Although OA is still the primary reason for THA with younger patients [4], they require THA for several other conditions such as osteonecrosis, rheumatoid arthritis, hip dysplasia, a slipped epiphysis, Perthes disease, congenital hip luxation, tumors and infection [1,4,6,7]. Regardless of age, symptoms for elective THA remain similar in that severe pain, limitation of range of motion (ROM) and functional limitations are primary reasons for THA [8]. THA provides substantial pain relief and resumption of many activities including sporting activities such as hiking, and cycling in younger patients [9]. After interviewing 405 patients post-THA, Mancuso et al. found that 87% of patients’ expectations (such as pain relief, ability to walk/stand or get out of a bed, perform daily activities and playing sport) had been fulfilled completely [10]. 

Primary aim of rehabilitation for recovery after THA is to optimize physical function, strength and pain. A systematic review based on 5 studies for 234 participants showed that 8–12 weeks of physiotherapy interventions improved hip abductor muscle strength, gait speed by 6 m/min (95% CI 1, 11) and gait cadence by 20 steps/min (95% CI 8, 32) [11,12]. Overall, rehabilitation programs for THA recovery have primarily been based on findings with older patients. Functional gains were reported in a small RCT study that found an eight-week strengthening program improved the 6 min walk distances in older adults at 12 and 24 weeks post-THA [12]. Improvements with hip muscle strength, postural stability and self-perceived function compared to the control group were seen at 4–12 months after surgery [13]. Studies examining specific exercise programs for younger patients are limited. A randomized control trial that examined the effect of a 4-week strengthening intervention after THA in patients 60 years or younger reported that maximal strength training program and an aerobic endurance training program are required for full recovery following THA in patients 60 years or younger [14].

Although substantial gains are seen with pain relief and functional improvement in the THA patient population [15], it is less clear that patients will become more mobile and return to an active lifestyle [16]. Physical activity does not innately improve after total joint replacement when pain is relieved and function improved [17,18]. Evidence and clinical guidelines generally support low impact physical activity for THA [19]; however, studies examining physical activity and function of younger patients following THA are lacking. A rehabilitation program specifically directed at younger patients recovering from THA is warranted given the special needs of this patient group regarding an active lifestyle and return to work.

The program evaluation examined the feasibility, safety and outcomes of a 6-week post-operative rehabilitation program compared to usual care in patients ≤60 years undergoing elective unilateral THA. 

## 2. Methods

### 2.1. Study Design and Patients

A consecutive sample of patients who received the THA program were matched on age and sex with concurrent controls who received usual care. Participants were recruited from a central intake hip and knee clinic which serves a population of approximately 1.5 million in Edmonton Alberta, Canada and surrounding regions. With over 35 surgeons and multi-disciplinary teams, approximately 3000 joint replacements are performed annually. Most, if not all, surgeons were high volume (>50 THA per year) who routinely used the antero-lateral or posterior-lateral hip approach. All patients followed a provincial standardized clinical pathway for acute-care management with a 3- to 4-day hospital stay. Post-operatively, patients attended clinic visits at 2, 6 and 12 weeks and one year by the orthopaedic surgeon and multi-disciplinary team. The home exercise program which is provided as usual care included range of motion activities, isometric hip strengthening exercises and walking with appropriate assistive walking devices.

Eligible participants (a) were 60 years of age or younger, (b) received a unilateral THA, (c) attended the 6- and 12-week follow-up after receiving an elective THA, (d) were able to communicate in English, and e) resided within the metropolitan area so they could attend the program. The exclusion criteria consisted of (a) bilateral THA, (b) hemiarthroplasties, (c) emergency THA, (d) primary diagnosis of developmental hip dysplasia, and (e) patients who had a systemic illness which limited their ability to participate (e.g., rheumatoid arthritis, cardiac involvement that precluded exercise). 

Upon providing written consent, patients allocated to the intervention group were assessed at their 6-week post-operative visit with the surgeon (baseline assessment) and then 6 weeks later when the interventional group had completed the rehabilitation program (follow-up assessment). Patients assigned to the control group were assessed at the same time points (Figure 1). The baseline assessment consisted of socio-demographic information (age, sex, education, marital status), medical/surgical information, pain and function (Hip Osteoarthritis Outcome Score, HOOS [20]) and participants’ satisfaction with THA. Physical activity was measured using the SenseWear Pro Armband^TM^ (SWA; Body Media, Inc., Pittsburgh, PA, USA) for 4 consecutive days immediately after consenting to participate at the 6-week clinic visit and then after the follow-up at the 12-week clinic visit. Using a standardized form, clinical charts were reviewed for medical and surgical information (chronic conditions, height/weight, surgical approach, complications). The follow-up assessment was at the 12-week post-operative surgical visit when the augmented rehabilitation program was completed. This study was approved by the University of Alberta’s Health Ethics Board. (PRO00051978).

### 2.2. Augmented Rehabilitation Program

The outpatient rehabilitation program involved 2 h of structured group exercise classes led by a physiotherapist over 6 weeks. Participants were required to attend twice a week for 12 sessions. Each session began with the 1 h aquatic session first, followed by a 1 h land-based exercise session. The aquatic sessions were performed in a therapeutic heated pool. Activities included walking, stair stepping, balance activities and strengthening exercises such as flutter kicks. The land-based session included a 5–10 min warm-up on stationary bicycles, followed by leg and core strengthening, stretching, balance activities in sitting and standing, closed kinetic chain exercises and gait re-training. Exercises were progressed to provide resistance as tolerated using either a theraband or small free-weights.

### 2.3. Usual Post-Operative Care 

A provincial care path required patients to attend a 6-week and 12-week post-operative surgical visit. At the 6-week visit, participants in the control group were matched for age (within a 5-year age range) and sex. They continued with usual care after their six-week surgeon’s visit which included a standardized home exercise of isometric strengthening and stretching exercises for the hip along with walking using the appropriate assistive walking device. 

### 2.4. Outcomes Measures

Feasibility was evaluated in terms of the retention of study participants, completion of the baseline and follow-up assessments, how well participants attended to the intervention components and the safety and tolerability of the intervention [21]. 

Physical Activity: A direct measure of daily PA was estimated from the daily energy expenditure calculated from the SenseWear Pro Armband^TM^ (SWA; Body Media, Inc., Pittsburgh, PA, USA). The SWA is a tri-axial accelerometer which integrates motion sensor data with the data from three sensors (heat flux, galvanic skin response and skin temperature) and combines these data with demographic characteristics (gender, height, weight, handiness and smoking status) to estimate energy expenditure using algorithms provided by the manufacturer [22,23,24,25]. The SWA has been shown to have good validity under both laboratory [22,26,27] and free-living conditions [28]. There is an error rate of 8% with an ICC of 0.8 for energy expenditure determined from the SWA and doubly labelled water [29]. 

The SWA was worn on the arm over the triceps for four consecutive days [30]. Participants were instructed to remove the device when bathing, showering or swimming. Valid days were defined as those days when the participant wore the SWA > 10 h during the waking time [31,32]. Once the SWA was returned to the lab, data were downloaded to generate the total time the device was worn, steps/minute and minute by minute energy expenditure and metabolic equivalent task (MET) intensity levels. One MET is the energy cost of resting quietly, often defined in terms of oxygen uptake as 3.5 mL/kg/min. Minute by minute data was averaged over all valid days to provide information on the number of steps/day, energy expenditure, stationary time and time spent in different activity intensities (i.e., light PA, moderate-vigorous PA (MVPA)).

Stationary time was defined as waking time with an energy expenditure ≤1.5 METs [33,34,35]. Light physical activity was defined as activities which required an energy expenditure of 1.6–2.9 METs (e.g., activities of daily living (ADL)). MVPA included activities with an energy expenditure ≥3.0 METs. For time spent in MVPA, recommendations suggested PA bouts of >10 min [36,37]. Data were inspected to determine both the total minutes of MVPA (MVPAall) and continuous MVPA occurring in >10 min bouts (i.e., MVPA10+). To count as an MVPA10+ the bout had to start with a minute above the MVPA cut-point (>3 METs) and last for >10 consecutive minutes with allowance for a maximum of two observations falling below the cut-point during the period (i.e., 8/10 min) [38]. Furthermore, we calculated the amount of time and energy spent in MVPA10+. The amount of energy spent in MVPAall was also presented as PA energy expenditure.

Hip Osteoarthritis Outcome Score (HOOS): The HOOS questionnaire is a self-reporting measure of joint-specific pain, stiffness, sports activity and health-related quality of life. It consists of 5 subscales: pain (10 items), symptoms (2 items), function in activities of daily living (ADL) (17 items), function in sport and recreation (Sport/Rec) (4 items), and hip-related quality of life (QoL) (4 items). Each subscale score is calculated independently by summing up all items in each subscale and then normalizing the total into the range of 0 to 100 [20]. A higher score reflects a better condition for the hip. The validity, reliability and responsiveness of HOOS had been evaluated in different studies [20,39,40]. Exercise therapy for hip OA impacts several outcomes including pain, function and QoL [41] which are relevant to the recovery of THA. HOOS contains adequate measurement qualities to evaluate patients with hip osteoarthritis or THA [40] and patients ≤66 years of age reported higher responsiveness in all five subscales compared to patients >66 years of age [20]. The minimal clinically important improvements in the HOOS subscales ranged from 17 to 24 points [42].

Participants’ satisfaction was assessed using a 5-item questionnaire asking if THA: could be recommended to their family, was worthwhile, was helpful, led to negative outcome or was dissatisfying [43]. All items were rated on a 5-point Likert scale ranging from strongly agree to strongly disagree.

### 2.5. Statistical Analysis

Summary descriptive were reported as mean and standard deviation (SD) for continuous variables, numbers and percentages for categorical variables. The number of participants who completed the study interventions and assessments were calculated. The number of adverse events were estimated. Participants were considered adherent to the intervention if they completed 80% or more of the intervention sessions. The between-group differences at baseline, follow-up and the absolute change (follow-up—baseline) were evaluated by using two independent samples t-tests. To evaluate the within-group differences, paired samples t-tests were used. Mann–Whitney U nonparametric test was used to compare the groups if data is not normally distributed. To explore whether the physical activity as measured by the SWA (step counts and MVPA bouts) were correlated with the self-reported HOOS subscale scores, Pearson correlation coefficients were examined and categorized as weak (0.10 to 0.39), moderate (0.40 to 0.69), strong (0.70 to 0.89) and very strong (0.90 to 1.00) [44]. Statistical analyses were conducted using SPSS version 24 (SPSS, Inc., Chicago, IL, USA).

## 3. Results

Of the 24 participants recruited, 14 participants received the augmented rehabilitation, and 10 participants were in the control group. The mean (SD) age of the intervention and the control groups were 51.9 (4.1) years and 54.5 (5.2) years, respectively. Although the two groups were comparable with respect to age, gender, marital status, living status and education (*p* > 0.05) (Table 1), the control group had higher body mass index (BMI) compared to the augmented rehabilitation group (mean difference 4.2 kg/m^2^; 95% CI 0.2, 8.3). The median number of comorbidities was significantly higher in the control group (4, IQR 3 to 5) than the augmented rehabilitation group (2, IQR 1 to 2) (*p* = 0.01). The most frequent conditions in both groups were chronic pain and mental health problems (Table 1). No post-operative surgical complications were reported during the study period. The length of stay for both groups was 3 to 4 days. 

All 24 participants completed the study including the baseline and follow-up assessments. No adverse events were reported by participants in the program over the 6 weeks. Thirteen of the fourteen intervention participants completed at least 80% of the 12 rehabilitation sessions. 

All participants were satisfied with the THA surgery procedure they had undergone. At the follow-up assessment, 12 (86%) patients in intervention group and 9 (90%) in control group strongly agreed with the statements that “I would recommend a hip replacement to my family if they needed care for hip arthritis” and “I feel the hip replacement surgery was worthwhile”. At follow-up assessment, 2 (20%) of the control group and 2 (14%) of the intervention group were dissatisfied with the functioning of the hip that received the replacement.

Table 2 summarizes the physical activity profile of the study groups at baseline and follow-up assessments. The mean wearing time of the SWA for the entire cohort was approximately 15 h/day over 4 consecutive days regardless of group. After 6 weeks of augmented rehabilitation, the intervention group took more steps/day compared to their baseline (mean difference = 2440 steps/day, 95% CI = 1678, 4712) (*p* < 0.05), although the daily step count of 10,000 steps was still below the recommended daily steps for adults [45]. In contrast, the mean daily step count for the control group showed no significant difference over time (mean difference = 573 steps/day, 95% CI = −1682, 2828) (*p* = 0.60). (Table 2).

The intervention group had a higher mean change of the number of MVPA bouts (6.2, SD = 20.7) than the control group (−9.2, SD = 16.4) (*p* = 0.05). The average time spent in each MVPA bout at baseline and follow-up was 15.1 ±4 min and 18.4 ± 10.6 min for the control group and 13.7 ± 4.1 and 17.2 ± 13.4 for the intervention group. The average time spent in the MVPA bouts was not statistically different between groups. No significant differences were seen between the intervention and control groups for any other parameters measured by the SWA.

Over time, the intervention group showed significant differences in all subscales of the HOOS, whereas the ADL subscale was only significant difference seen with the control group (Table 3). At baseline, the control group (78.5 ± 18.4,) had less symptoms (HOOS Symptom) than the intervention group (63.9 ± 14.3) (*p* = 0.04). The mean change in the symptoms and pain HOOS subscale were significantly higher in the intervention compared to the control group. At the follow-up assessment, the sport/recreation subscale was significantly higher in the intervention group (79.9 ± 16.7) compared to the control group (57.5 ± 26.0) (*p* = 0.02) (Table 3). Only the intervention group showed a clinical important change compared to their baseline in ADL, QoL and sport/rec; however, there was no clinical important change in the control group [46]. While most correlations between the HOOS and daily step count or the number of weekly MVPA bouts were weak (r = 0.1–0.39), moderate correlations (r = 0.4–0.69) were seen between the HOOS-QoL subscale and step counts (r = 0.49), and number of MVPA bouts (r = 0.53).

## 4. Discussion

The augmented rehabilitation program was feasible, acceptable and safe in patients ≤60 years undergoing elective unilateral THA. In spite of the small sample, the augmented rehabilitation group had greater mean changes in step count and the number of MVPA bouts of activity, and less reported HOOS symptoms and pain subscales than usual care. Although physiotherapy-based exercise programs have been found to be effective [47], only one small clinical study specifically examined community rehabilitation program for patients 65 years or younger during long-term recovery [48]. They did not examine physical activity yet reported no group differences with gait parameters such as walking speed [48].

Other studies that have examined step count in younger patients recovering from THA reported fewer daily steps at 6 months after surgery [49]. In comparison to our intervention group, Fujita et al. [50], reported lower step counts (5657 steps), light activity (107 min/day) and MVPA (17 min/day) at 6 months after THA). The differences in step counts and MVPA bouts could be due to the different methods of measuring objective physical activity, such as pedometers [50], mobile step-tracking application [49] or accelerometer. 

Physical activity guidelines for adults aged 18–64 years include at least 150 min of MVPA per week to achieve health benefits [51,52,53]. The average time patients in the intervention group spent on MVPA daily was approximately 176.2 min/week and they met the recommendations of physical activity. Compared to the control group, the intervention group has more MVPA bouts per week (21.0 versus 11.3) at the follow-up assessment than the control group. After accounting for baseline differences, there was a statistically significant differences of MVPA bouts between groups. These between-group differences appear to be clinically important and show the potential effectiveness of the intervention.

The positive effects of the augmented rehabilitation we found may be attributed to a few features of the program such as structured classes led by physiotherapists, high intensity activities, and available equipment. A systematic review found that the center-based exercise rehabilitation is more effective than the home-based exercise for THA patients’ recovery because of the higher training intensity and access to professional supervision, specialised equipment and facilities in center-based exercise [54]. Hydrotherapy for hip OA has positive effects to reduce pain and improve physical function and quality of life [55,56]. Aquatic exercise offloads weight bearing joints and may allow people with hip pain, swelling and leg weakness to exercise successfully when this is not possible on land [57,58]. Lastly, as the PA guidelines recommended, it is beneficial to add muscle and bone strengthening activities to the exercise regimen for people aged 60 years or younger [51,52,53].

More recently, physical activity has become another outcome. Others have shown weak to moderate correlation between PA using an accelerometer and self-reported PA questionnaires in people with hip OA and post-THA [59]. We found similar findings in that the correlation between the bouts of MVPA and step count did not correlate with self-reported pain and function, yet moderate correlations were seen with the HOOS QoL and step counts and MVPA bouts. This correlation confirms the results of previous population-based studies that showed association of health-related quality of life and physical activities [60,61]. Emerging evidence supports the use of digital devices that provide real-world data to monitor physical activity and sedentary time in people with OA [62] rather than relying on recall of physical activity with self-reported PA measures.

By offering a community-based group rehabilitation program for THA, structured long-term rehabilitation is provided for recovery. A concern of rehabilitation concerns earlier hospital discharges for THA that are attributable to advances in surgical and prosthetic technologies. The ramification of an earlier hospital discharge is that the recovery time with a specialized care team and active rehabilitation has been limited [63]. At discharge, patients receive a guide for continuing strengthening, stretching and ambulation exercises; however, they are expected to complete this exercise program independently. In a prospective study, it was reported that many patients would have liked to spend more time with their physiotherapists which could be due to insecurities or lack of confidence of the patient in performing activities alone [63]. The proposed post-operative rehabilitation program provides additional contact and supervised exercise time with physiotherapists in a community setting. 

Our findings should be considered within context of this feasibility study design in that the program was acceptable as reflected by attendance; retention, and no adverse events, or complications reported. This study addresses an understudied area of post-THA rehabilitation of patients 60 years or younger. A strength of this study was not only including self-report measure of THA (HOOS) and objective measure of PA, but also examining the correlation between these two measures. We showed the feasibility of implementing an augmented rehabilitation program (including land and water-based exercise) in THA patients who are 60 years or younger and support a definitive trial.

Although this study was under-powered, the reported change seen in outcomes will be valuable in sample size calculations for larger trials. Optimal duration and frequency of the exercise sessions have yet to be determined and warrants further investigations. Long-term sustainability of the program is unknown, and recommendations for a 6-month follow-up may be warranted given that other post-operative rehabilitation programs for THA have reported sustained change for 6 months [49].

## 5. Conclusions

The study intervention and assessments were feasible and safe in patients ≤60 years undergoing elective unilateral THA. The post-operative augmented rehabilitation program for younger patients with THA may lead to immediate increase in step counts, MVPA bouts, self-reported function and pain reduction. The findings provide insights into the pragmatic issues of implementing a land and water-based rehabilitation program for younger patients. Further study with a large sample and long-term follow-up is warranted to assess the long-term effects of our augmented rehabilitation program.

## Figures and Tables

**Figure 1 healthcare-10-01274-f001:**
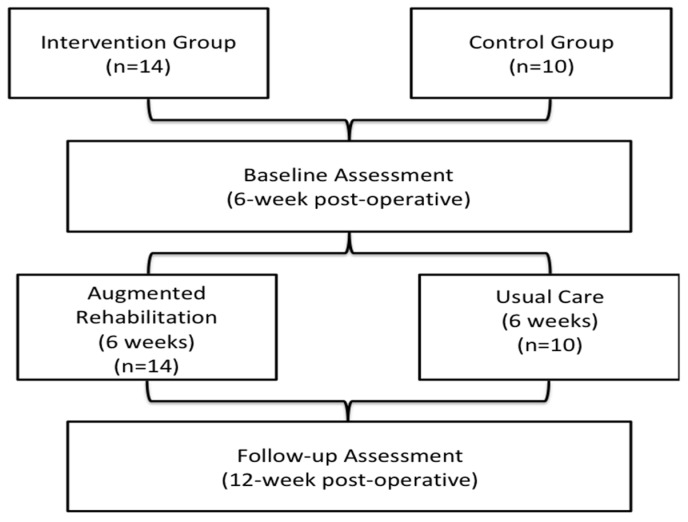
Study Flow chart.

**Table 1 healthcare-10-01274-t001:** Baseline characteristics of participants.

	Intervention (*n* = 14)	Control (*n* = 10)	*p* Value
**Demographics**			
Age (years), mean ± SD	51.9 ± 4.1	54.5 ± 5.2	0.184
Female, n (%)	5 (35)	3 (30)	0.836
Employment Status, n (%)			
Employed	10 (71%)	6 (60)	0.752
Unemployed	1 (7%)	0 (0)	0.398
Disability	2 (14%)	1 (10)	0.822
Retired	1 (7%)	3 (30)	0.229
Marital Status, n (%)			
Married/ common-law	12 (86)	9 (90)	0.999
Single/widowed/divorced	2 (14)	1 (10)	0.822
Education (years of school), mean ± SD	14 ± 2.3	14.1 ± 2.5	0.920
Living Status, living with, n (%)			
Alone	1 (7)	0	0.398
Spouse and/or others	13 (93)	10 (100)	0.999
**Medical**			
BMI (kg/m^2^) mean ± SD	27 ± 4.1	31.2 ± 5	0.034
Number of co-morbidity/person, median (25%ile, 75%ile)	2 (1, 2)	4 (3, 5)	0.010
3 Most Prevalent Comorbidities			
Chronic Pain, n (%)	8 (57)	6 (60)	0.999
Mental Health Problem, n (%)	4 (29)	5 (50)	0.421
High Blood Pressure, n (%)	4 (29)	3 (30)	0.999
Ambulation Status, n (%)			
Distance walk <1 block	1 (7)	0 (0)	0.398
Distance walk 1–5 blocks	4 (29)	6 (60)	0.263
Distance walk 6–10 blocks	4 (29)	1 (10)	0.376
Distance walk unlimited	5 (36)	3 (30)	0.836
**Surgical characteristics**			
Type of Arthroplasty, n (%)			
Non-cemented	11 (79%)	8 (80)	0.999
Hybrid	2 (14%)	2 (20)	0.999
Cemented	1 (7%)	0 (0)	0.398
Surgical Approach, n (%)			
Antero-lateral hip	2 (14)	1 (10)	0.822
Posterior-lateral hip	9 (65)	4 (40)	0.448
Unknown	3 (21)	5 (50)	0.261

*Abbreviations:* SD—standard deviation; BMI—body mass index.

**Table 2 healthcare-10-01274-t002:** Comparisons of physical activity measures within groups at baseline and follow-up.

	Intervention (*n* = 14) Mean ± SD			Control (*n* = 10) Mean ± SD		
Physical activity Measures	Baseline	Follow-up	*p*-value	Baseline	Follow-up	*p*-value
Waking wearing time (Hrs/day)	14.9± 1.5	14.7± 1.4	0.658	15.6 ± 1.4	15.3 ± 1.2	0.782
Sleeping time (Hrs/day)	5.8 ± 1.9	4.7 ± 1.1	0.058	6.0 ± 1.0	5.2 ± 1.0	0.050
Step (counts/day)	4156 ± 2460	6596 ± 3325	0.005	5282 ± 2720	5855 ± 2031	0.439
Stationary time (Hrs/day)	10.8 ± 2.1	10.1 ± 2.1	0.120	10.5 ± 2.3	10.5 ± 1.8	0.906
Mild Activity (Hrs/day)	2.9 ± 1.2	3.1 ± 0.9	0.496	3.6 ± 1	3.6 ± 1.4	0.889
MVPA (Hrs/day)	1.1 ± 1	1.6 ± 1.4	0.085	1.5 ± 1.1	1.3 ± 0.7	0.463
Weekly Number of bouts ≥10 min at MVPA	14.8 ± 14.6	21.0 ± 29.2	0.286	20.4 ± 16.5	11.3 ± 8.6	0.110
Duration of MVPA bout	13.0 ± 4.1	16.9 ± 13.4	0.402	15.1 ± 3.9	18.4 ± 10.6	0.100
DEE (kcal/day)	2184 ± 470	2227 ± 446	0.531	2686 ± 768	2546 ± 714	0.316
All PAEE > 3 METs (kcal/day)	322 ± 252	445 ± 261	0.073	557 ± 474	477 ± 285	0.506

*Abbreviations:* SD: Standard deviation, MVPA: Moderate-to-vigorous physical activity, EE: energy expenditure, DEE: Daily energy expenditure, PAEE: Physical activity energy expenditure. Bold *p*-values indicate statistical significance

**Table 3 healthcare-10-01274-t003:** Comparisons of HOOS scores within groups and between groups.

	Intervention (*n* = 14) Mean ± SD			Control (*n* = 10) Mean ± SD			*p*-Value for Group Effect *		
HOOS Scores	Baseline	Follow-Up	*p*-Value	Baseline	Follow-Up	*p*-Value	Baseline	Follow-Up	Mean Change
Pain	74.2 ± 20.2	91.7 ± 8.6	**0.005**	80.6 ± 18	83.6 ± 14.5	0.387	0.436	0.141	**0.029**
Symptom	63.9 ± 14.3	82.9 ± 9.7	**<0.001**	78.5 ± 18.4	84.5 ± 10.9	0.250	**0.040**	0.305	**0.037**
ADL	73.1 ± 16.3	89.1 ± 7.7	**0.003**	74.3 ± 18.5	83.1 ± 13.1	**0.022**	0.873	0.172	0.207
Sport	42.0 ± 26.1	79.9 ± 16.7	**<0.001**	43.8 ± 29.2	57.5 ± 26.0	0.312	0.876	**0.017**	0.117
Hip QoL	42.9 ± 19.0	63.4 ± 17.7	**0.007**	53.8 ± 22.5	61.3 ± 18.6	0.161	0.212	0.777	0.122

*Abbreviations:* HOOS—Hip Osteoarthritis Outcome Score; SD—standard deviation; CI—Confidence Interval; ADL—Activities of Daily Living; Sport—Sport and Recreation Function; Hip QoL—Hip-Related Quality of Life. *** Between group comparisons examined difference between the intervention and control groups. Bold *p*-values indicate statistical significance

## Data Availability

Not applicable.

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
