# Peer review of "Augmented Rehabilitation Program for Patients 60 Years and Younger Following Total Hip Arthroplasty—Feasibility Study"

_healthcare, 2022, doi:10.3390/healthcare10071274_

Round 1

Reviewer 1 Report

The manuscript titled “Augmented Rehabilitation Program for Young Patients Following Total Hip Arthroplasty- feasibility study” investigated the administration of an augmented physical activity program in a group of THA patients, comparing the results with a group undergoing the classic rehabilitation procedure. This method seems promising, even if several concerns have to be solved to ensure the publication of this manuscript.

The whole manuscript is confusing and not very concise. Many terms are repeated several times (such as the PA collection through the wristband), making reading difficult to follow. Authors should simplify it and point out their findings better.

Title

Young patients – age < 60 years cannot be defined as young patients; the title is misleading. I suggest removing the term “young” from the paper (individuals aged 50-55 years are not young adults).

Abstract

Revise the headings

Background is not the background of the manuscript, it is the aim. Rewrite it.

L 13 why are authors using square brackets?

L17-19 authors are reporting the step/day as a comparative value to show the outcomes, but before this “values” has been not cited.

L21-22 the conclusions are not conlusions. Authors are just saying that their method is worth, nothing else. Conclusions should summarize the validity of this method compared with the classic one.

Introduction

L68 authors have to add here the abbreviation of physical activity instead of L78

L75-77 Why are authors adding cit 25 to their aim?

L77-81 The aim is too confusing, I suggest rewriting it.

Methods

L85-90 The endorsement to the Edmonton clinic is not required, remove it.

L108 “The baseline interview assessment … physical activity (SenseWear Pro Armband (SWA)” did the authors measure the PA at baseline?

L119 “2.2. Augmented Rehabilitation Program” How should readers be stimulated to read or study this paper if the rehab program is not described? Furthermore, the authors said that the main aim is to evaluate the feasibility of this program—which program? It would be interesting adding a table or graphics that show the augmented rehab program.

L136 remove (PA) abbreviation

L184 “Statistical Analysis” something is unclear to me. Authors used t-test to evaluate the differences between baseline and follow-up; then used the t-test to compare the experimental and the control group results for which variable? This “multiple t-test” seems to be a violation.

Results

Results are a bit messy. I’d suggest revising them.

To me, table 3 is not clear to understand. What is the “p-value between-groups comparisons” assessing?

Discussion

L275-76 Seems like authors are defending their results because “people are sedentary and sedentary behavior is difficult to change”. This is not what and author is called to do to argue the results.

L277 Why are authors referring to Canadian PA guidelines and not WHO or ACSM guidelines?

________________________________

Furthermore, I would suggest you some interesting papers which may give you some valuable food for thought to include in your work eventually:

1)     Aquatic exercise for the treatment of knee and hip osteoarthritis (https://pubmed.ncbi.nlm.nih.gov/27007113/) suitable for L291-298

2)     Relative Efficacy of Different Exercises for Pain, Function, Performance and Quality of Life in Knee and Hip Osteoarthritis: Systematic Review and Network Meta-Analysis (https://pubmed.ncbi.nlm.nih.gov/30830561/) suitable for L168-179

3)     Exploiting real-world data to monitor physical activity in patients with osteoarthritis: the opportunity of digital epidemiology
(https://pubmed.ncbi.nlm.nih.gov/35252602/) suitable for L299-306

The first one provides exciting information about the aquatic exercises, and since it is a part of your rehab program, you may find interesting points.

The second is a valuable summary of the existing papers about hip OA so that you may find good suggestions about the QoL.

The third one investigated using digital devices (smartwatches ecc) to evaluate the steps/day in OA patients. It may fit well the content of your discussion.

Minor checks:

Check the terms “table” and similar because they are formatted wrong.

L266 add the citation after the name of the author.

Reviewer 2 Report

Thank you very much for presenting your method of rehabilitation that gives excellent results in patients under 60 years of age who underwent standby unilateral THA.

Why was the rehabilitation in this study a 2-hour intervention twice a week? Do you have any previous data showing that once a week or one hour of training per session is not enough? Also, is it possible that training 3 or more times per week would have been more effective? Any data or opinions you may have would be helpful to our readers in their own rehabilitation efforts with their own patients.

I am curious about the significant differences in BMI and number of comorbidities between the intervention group and control groups. If possible, please explain the impact of this condition on the results.

It would also be helpful to the reader if you could describe the specifics of this training in more detail.

Finally, thank you very much for the opportunity to review such a very valuable paper.

Round 2

Reviewer 1 Report

The authors addressed all the previous comments. Just a small issue has to be fixed before suggesting its publication.

Authors say "We are uncertain about deleting the ‘Background’ heading since the journal headings require the heading of ‘Background’." but in the Healthcare's template it's reported literally -> We strongly encourage authors to use the following style of structured abstracts, but without headings"

Author Response

Thank you for the clarification. We have deleted the 'Background' heading in the Abstract.